# Exploring women's decisions of where to give birth in the Peruvian Amazon; why do women continue to give birth at home? A qualitative study

Esme Gardiner[1]*, Jo Freda Lai[1], Divya Khanna[1], Graciella Meza[2], Gilles de Wildt[1], Beck Taylor[3]

**1** College of Medical and Dental Sciences, University of Birmingham, Birmingham, United Kingdom, **2** Facultad de Medicina Human, Universidad Nacional de la Amazonía Peruana, Iquitos, Peru, **3** Institute of Applied Health Research, University of Birmingham, Birmingham, United Kingdom

* esmegardiner@btinternet.com

**Data Availability Statement:** This study is based on a dataset of 25 qualitative interview transcripts. However, the authors did not seek ethical

## Abstract

### Background

Despite improvements in maternal mortality globally, hundreds of women continue to die daily. The World Health Organisation therefore advises all women in low-and-middle income countries to give birth in healthcare facilities. Barriers to seeking intrapartum care have been described in Thaddeus and Maine's Three Delays Model, however these decisions are complex and often unique to different settings. Loreto, a rural province in Peru has one of the highest homebirth rates in the country at 31.8%. The aim of this study was to explore facilitators and barriers to facility births and explore women's experiences of intrapartum care in Amazonian Peru.

### Methods

Through purposive sampling, postnatal women were recruited for semi-structured interviews (n = 25). Interviews were transcribed verbatim and thematically analysed. A combination of deductive and inductive coding was used. Analytical triangulation was undertaken, and data saturation was used to determine when no further interviews were necessary.

### Results

Five themes were generated from the data: 1) Financial barriers; 2) Accessing care; 3) Fear of healthcare facilities; 4) Importance of seeking care and 5) Comfort and traditions of home. Generally, participants realised the importance of seeking skilled care however barriers persisted, across all areas of the Three Delays Model. Barriers identified included fear of healthcare facilities and interventions, direct and indirect costs, continuation of daily activities, distance and availability of transport. Women who delivered in healthcare facilities had mixed experiences, many reporting good attention, however a selection experienced poor treatment including abusive behaviour.

permission from the participants, nor the ethics committee, for the data to be used for anything other than this particular research study. The authors therefore do not have explicit permission for data sharing, re-analysis nor future studies and so would be inappropriate and unethical to make them available in the public domain. Furthermore, the data contains potentially identifying patient information. However, qualified individuals can direct queries by contacting Dr Ruth Riley (r. riley@bham.ac.uk) - chair of the University of Birmingham BMedSci Intercalation Internal Ethics Review Committee.

**Funding:** This work was supported by the University of Birmingham and an intercalated bursary award from the Topham Bursary Fund (EG). Support was also provided by the National Institute for Health Research (NIHR) Applied Research Centre (ARC) West Midlands (BT). The views expressed are those of the author and not necessarily those of the NIHR or the Department of Health and Social Care. The funders had no role in the study design, data collection and analysis, decision to publish or preparation of the manuscript.

**Competing interests:** The authors have declared that no competing interests exist.

**Abbreviations:** COREQ, consolidated criteria for reporting qualitative research; EsSalud, Social Health Insurance; IPC, Intrapartum Care; LMIC, Lower-middle Income Country; MDGs, Millennium Development Goals; MMR, Maternal Mortality Rate; SBA, Skilled Birth Attendant; SDGs, Sustainable Development Goals; SIS, Seguro Integral de Salud; TDM, Three Delays Model; WHO, World Health Organisation.

## Conclusion

Despite free care, women continue to face barriers seeking obstetric care in Amazonian Peru, including fear of hospitals, cost and availability of transport. However, women accessing care do not always receive positive care experiences highlighting implications for changes in accessibility and provision of care. Minimising these barriers is critical to improve maternal and neonatal outcomes in rural Peru.

## Introduction

Despite advances in recent years, maternal mortality remains a major global issue. The maternal mortality rate (MMR) is defined as the number of maternal deaths per 100,000 live births, caused by conditions related to or aggravated by pregnancy [1]. Despite aspirations within the Millennium Development Goals (MDGs) to reduce MMR by 75% by 2015, only a 38% reduction was achieved, and hundreds of women continue to die daily [2–4]. Many of these deaths are preventable, highlighting the importance of high quality antenatal and intrapartum care (IPC) [5]. The World Health Organisation (WHO) state that all women have a right to access high quality care during pregnancy and parturition, however less than half of births in low and middle-income countries (LMIC) are attended by healthcare professionals [6, 7]. Preventing these deaths is a key priority for the WHO, specifically addressing inequalities in access to and quality of maternity services [6, 8].

The Sustainable Development Goals (SDGs) aim to reduce MMR to less than 70 in every country and achieve universal access to skilled birth attendants (SBAs) [9]. SBAs are defined as "accredited health professionals who has been educated and trained to proficiency in the skills needed to manage an uncomplicated pregnancy and childbirth and identify, manage and refer complications" [10]. Clear evidence links the presence of SBAs with improved maternal outcomes [11–13]. Therefore, the WHO advises women to give birth in healthcare facilities, allowing access to SBAs and timely referrals if required [10].

The Three Delays Model (TDM), described by Thaddeus and Maine, outlines barriers to accessing maternity care in LMICs (Fig 1) [14]. These range from deciding to seek care to receiving adequate care. Understanding factors which prevent or facilitate women attending obstetric care is pivotal to achieve universal access to SBAs and prevent maternal deaths.

With an MMR of 68, Peru has achieved the SDGs, however when compared to other upper-middle income countries, Peru's statistic is 50% higher [15, 16]. The WHO and Pan American Health Organisation recently announced an aim to further reduce MMR to 65 by 2021. 93.1% of all births in Peru are attended by an SBA [9]. However, deliveries outside of healthcare settings, and therefore often without the presence of SBAs, are up to 9.3 times more common in rural areas [17]. Loreto is the largest region of Peru, located within the Amazon basin and has one of the highest rates of MMRs [119.5] and homebirths in the country at 31.8% [18, 19].

As far as the authors are aware, there are no publications in English, or with an English language abstract, about why women continue to give birth at home in Peru or Amazonia. Most of the global literature is based in Sub-Saharan Africa where MMRs are highest [4]. Barriers documented include lack of education about childbirth, cultural disparities with healthcare, distance, quick onset of labour and poor treatment from HCPs [20–32]. Cost is also often cited as a barrier for accessing IPC in many LMICs. Since 2002, Peru has had a Decentralised National Healthcare System to help achieve comprehensive healthcare for all the population

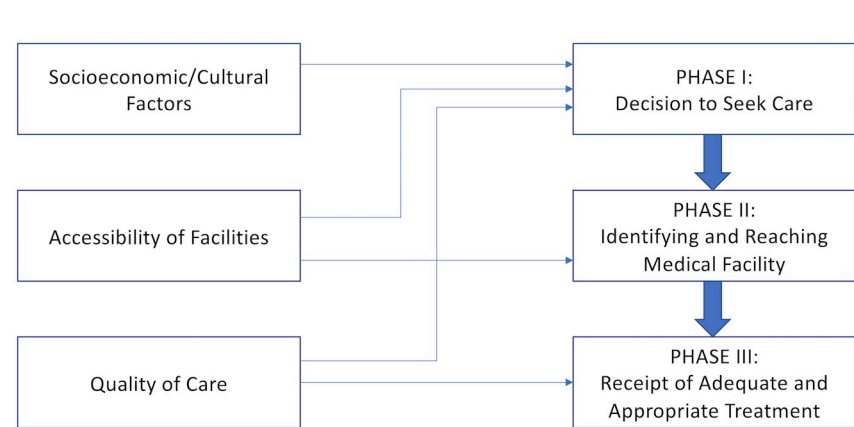

**Fig 1. The Three Delays Model: Thaddeus and Maine (1994).**

[33]. The Ministry of Health provides free antenatal and IPC through the government funded *Seguro Integral de Salud* (SIS) programme for the poorest of the population and *Social Health Insurance* (EsSalud) for employees [34].

Marsland et al., conducted a study about women's perceptions of antenatal and IPC in Loreto; women reported free appointments facilitated antenatal care attendance and IPC was often poor, including discontinuity of care [35]. Westgard et al., also reported that women in rural Peru understood the importance of seeking obstetric care, particularly with complicated pregnancies [36]. However, no research has been conducted in Peru or Amazonia as to why women give birth at home. Literature from other countries has shown that these decisions are complex and multi-factorial, so in-depth qualitative research is required to help address barriers and improve uptake of obstetric care [37].

The aim of this study was to research why women in Loreto choose to give birth at home or in healthcare facilitates. This involved exploring facilitators and barriers to health-facility births, to understand the multi-factorial reasons for the low uptake of IPC. Additionally, this study explored women's experiences of IPC in healthcare facilities to highlight possible improvements. Recent publication by the Regional Government of Loreto indicated key areas for health research, including exploring sociocultural factors that influence location of childbirth and reasons for delays in accessing obstetric care [38].

## Methods

### Ethical approval

Written ethical approval was provided locally by the Regional Directorate of Health, Loreto (385-2019-GRL-DRSL/30.09.01) and The University of Birmingham–BMedSc Population Sciences and Humanities Internal Research Ethics Committee (IREC2019/1548737). All participants provided written, informed consent prior to conducting the interviews.

### Methodological approach

This paper details an interpretive, descriptive qualitative study aligned with the *consolidated criteria for reporting qualitative research* (COREQ) checklist (S1 File) [39]. Qualitative methods were chosen to allow participants to express beliefs, feelings and motivations that underpin

their behaviour [40]. Semi-structured interviews were chosen over focus groups, to allow for potentially sensitive content [41, 42].

## Setting

Loreto, a large rural province in north-eastern Peru has one of the highest rates of homebirths in the country at 31.8% [17]. Despite Peru's economic development, Loreto continues to experience poverty and poor healthcare provision [43]. Iquitos, the capital of Loreto, and neighbouring villages are surrounded by the Amazon, Nanay and Itaya rivers; making them inaccessible by road [44, 45]. The majority of the population describe themselves as indigenous or descendants of indigenous people and the predominant language is Spanish (92.6%) [46]. The literacy rate of the population is 92.6% and over 75% of the population is Roman Catholic [47].

## Participant characteristics

Women who had given birth at home and in healthcare facilities were included in the study to allow exploration of both facilitators and barriers to facility births. The eligibility criteria are detailed in Table 1. Participant demographics were collected through a questionnaire to contextualise findings and allow reporting of characteristics (Table 2). Birth location was reported from participants' last pregnancy. However, an additional two women had previous experience of homebirths.

## Recruitment and sampling

Participants were recruited from January to February 2020, through a purposive method in primary healthcare centres within Iquitos and a door-to-door approach. The healthcare centres, Centro-de-Salud San Juan and Centro-de-Salud Moronacocha, serve the San Juan Baustica and Iquitos regions respectively. These healthcare centres or 'postas', provide an array of services including general medicine, paediatrics, dentistry and maternity care. All care is provided free of charge through the SIS. Participants were also recruited in six villages and communities within three hours of Iquitos, accessible by either road or boat (Fig 2). Due to time and financial restraints of the project, a pragmatic approach was taken, and other villages could not be visited. A visualisation of the recruitment process is detailed in S2 File.

## Data collection

Data was collected through face-to-face semi-structured interviews (n = 25, average length 34 minutes), including one pilot interview. Interviews were structured with a topic guide (S3 File), developed for this study from the TDM, other studies and discussion between authors [14, 25, 35, 49]. The topic guide was refined iteratively and used to ensure consistency between interviews whilst allowing the researcher flexibility to explore topics [50, 51]. Interviews were transcribed concurrently with data collection to allow a constant comparative approach.

**Table 1. Eligibility criteria.**

| Inclusion Criteria | Exclusion Criteria |
|---|---|
| Women who had given birth within the previous 18 months | Women who did not have capacity to consent |
| Speak English or Spanish as their first language | Serious illness or death of the new-born |
| Permanent resident in Loreto | |
| Over the age of 18 | |

**Table 2. Socioeconomic characteristics of participants.**

| Characteristic | Number of participants (%) |
|---|---|
| **Location of most recent birth** | |
| Hospital | 12 (48) |
| Posta | 4 (16) |
| EsSalud | 3 (12) |
| Home | 6 (24) |
| **Age** | |
| 18–24 | 8 (32) |
| 25–29 | 8 (32) |
| 30–34 | 6 (24) |
| ≥35 | 3 (12) |
| **Parity** | |
| 1 | 4 (16) |
| 2 | 8 (32) |
| 3 | 8 (32) |
| 4 | 3 (12) |
| ≥5 | 2 (8) |
| **Ethnicity** | |
| Mestizo | 19 (76) |
| Other | 6 (24) |
| **Education completed** | |
| No schooling complete | 2 (8) |
| Primary | 9 (36) |
| Secondary | 10 (40) |
| Further education | 4 (16) |
| **Occupation** | |
| Housewife | 20 (80) |
| Student | 2 (8) |
| Other | 3 (12) |

All interviews were conducted in Spanish by EG via an experienced interpreter, independently recruited from the Universidad Nacional de la Amazonia Peruana. Neither the researcher nor interpreter were involved in the care of participants and the interpreter was asked to sign a confidentiality agreement. All interviews were digitally recorded and supplemented with reflective field notes.

## Data analysis

All interviews were transcribed verbatim into the English and Spanish segments by the principal research and interpreter respectively. Hybrid verbatim was chosen to ensure narrative flow whilst including fillers and interjections [52]. The Spanish was translated and compared against the English to assess the quality of translation and to ensure all interview data was included. Following the interviews, reflections and a list of interview topics were made. After 23 interviews, it was recognised that no additional issues were being discussed. A further two interviews were conducted, and it was decided by the research team that analytical saturation had been achieved [53, 54]. Recruitment and further data collection was then ceased.

Data was managed using NVIVO12 and thematically analysed following the 6-step approach described by Braun and Clarke [41]. A combination of deductive and inductive

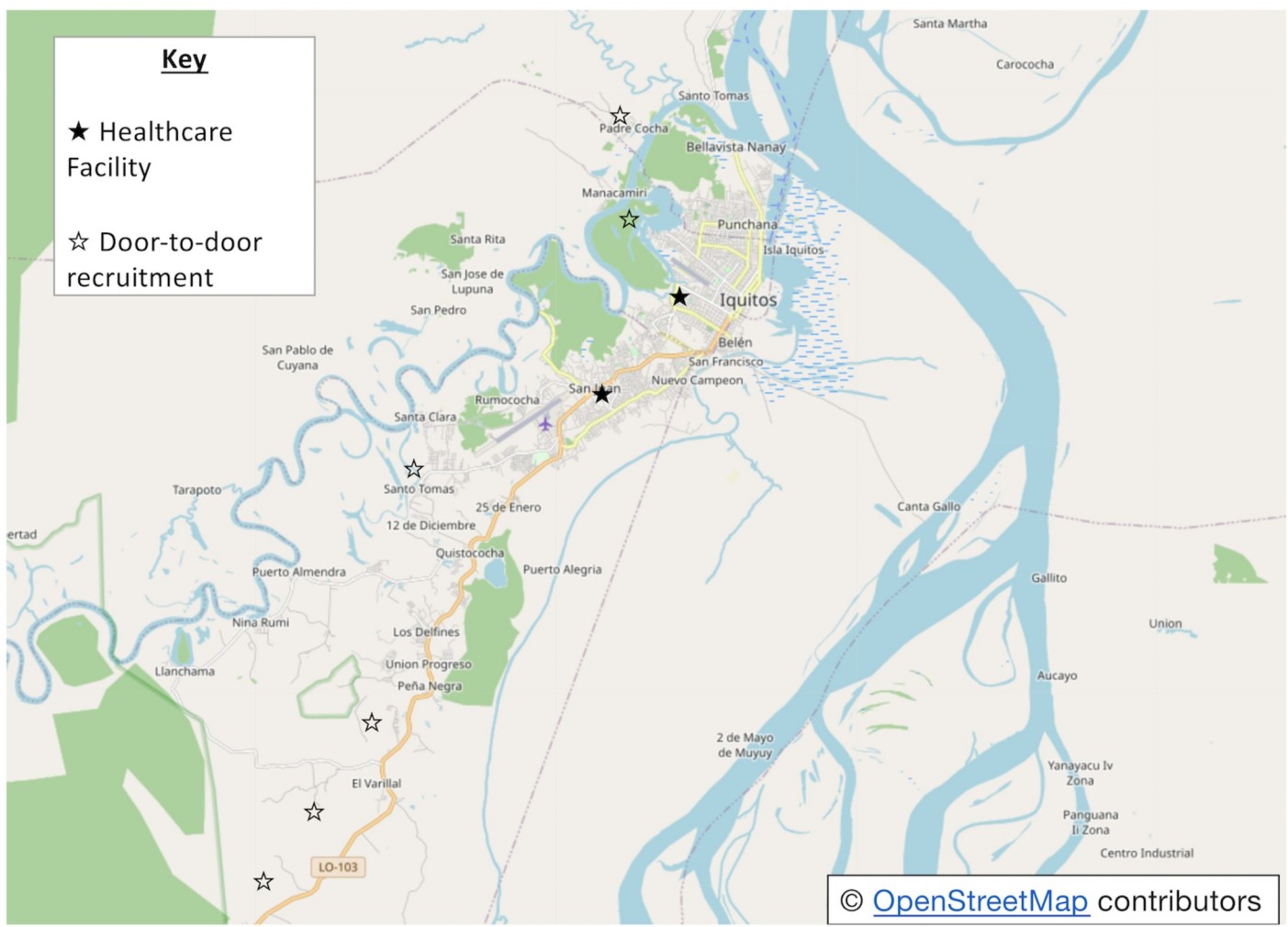

**Fig 2. Map of recruitment sites including primary healthcare facilities and neighbourhoods.** Adaptation of image from Google Maps [48].

coding was used; with concepts from the TDM labelled deductively whilst identifying other elements inductively [55]. A set of four transcripts were coded by two research assistants and compared to the principal researchers to ensure rigour [56]. Following the deductive and inductive rounds of coding, codes were arranged into themes and reviewed using a constant comparative approach [57]. The final themes were refined and named through discussion between authors.

## Results

Five key themes were developed from the data (Table 3). Extracts of data are presented alongside participant details; some quotations have been adapted to ensure participant anonymity. A further breakdown of the themes is available (S4 File).

### Theme 1—Financial barriers

Healthcare in Peru is supported by two health insurance schemes which provide free antenatal and IPC. Since the introduction of both schemes, rates of births with SBAs have increased

**Table 3. Themes and subthemes.**

| Themes | Subthemes |
| --- | --- |
| Financial Barriers | Insurance |
| | Additional costs |
| | Poverty |
| Accessing care | Transport |
| | Distance |
| | Nature of labour–speed and pain of labour |
| Fear of Healthcare Facilities | Fear of hospital |
| | Fear of intervention |
| | Discomfort with hospital care |
| Importance of seeking care | Realisation of the importance of care |
| | Poor education |
| | Healthcare practitioner's advice |
| Comfort and Traditions of Home | Comfort of home |
| | Care available at home |
| | Continuation of daily activities |
| | Avoiding hospital experiences |
| | Autonomy |

[58]. Throughout the interviews, women suggested the availability of these systems encouraged facility births. However, for a handful of women, despite having access to either the SIS or EsSalud, other barriers arose, preventing women accessing this care; including indirect costs of facility-based deliveries.

*"I didn't go to the hospital, I didn't have any money. I mean, although in the hospital they don't ask for money. . . I didn't have enough for the ticket to get out of here."*– 18yrs, Home

Furthermore, many of the women discussed insufficiencies in the system whereby additional costs were incurred, notably for medication.

*"Because in the hospital not. . . not everything is covered by SIS. . . SIS only covers the simple, most necessary thing you need. The doctors asked us for the medicine. . . we had to cover it from our pocket."*– 21yrs, Hospital

These additional costs can be even greater when complications arise during labour:

*"Suddenly you get a. . . a few stitches or you get another disease and. . . that isn't covered by the SIS"*– 39yrs, Home

The high rates of poverty in Loreto were also evident [19]. This combined with the unpredictable costs of IPC influenced women's decision to choose to give birth at home without the assistance of SBAs.

*"I thought. . . that it costs a lot in the hospital, and at home I am not going to spend. . . I thought about it, the hospital is going to take a lot of money from me"*– 36yrs, Home

Despite the free IPC available to many women in Loreto, participants experienced and feared additional costs. This has resulted in financial barriers to accessing IPC, ultimately resulting in a few women deciding to give birth at home.

## Theme 2—Accessing care

Problems accessing care were discussed by many, including distance and cost of transport, as previously mentioned. In addition, the availability of transport influenced women's decision of where to give birth. Generally, women living within Iquitos reported good availability of 'motokars'. However, connectivity fell within a couple of kilometres of the city. In rural villages, women relied on the river and public boats for transport. This caused challenges to accessing care, made worse during the dry seasons.

*"Going in the month of October when the river was dry to get out of here is a little difficult, to go to the city. That's why I stopped going to the city, because the river is far away."*– 44yrs, Posta

Additionally, the river transport system made accessing IPC particularly difficult with the onset of labour at night when public boats do not run.

*"Only the boats work from. . . in the morning. From 4 am in the morning until 5 or 6 pm in the afternoon, from then on there are no boats. . . At night, there is no longer any available"*– 44yrs, Posta

Distance and travel time to healthcare facilities also influenced many women's decisions; one woman discussed the fear of delivering en route.

*"I told them the [labour] pains were coming quickly. . . and the ambulance won't come soon. It's more dangerous to deliver my baby on the road and so I have my baby here"*– 18yrs, Home

To enable a facility-based delivery, two women who lived in rural villages chose to travel to Iquitos prior to their delivery dates to ensure reaching healthcare facilities.

*"That's why I stayed there in Iquitos so I could get to the post faster."*– 44yrs, Posta

The speed and pain of labour also influenced many women's ability to access care.

*"I got a pain. . . I couldn't walk anymore so I had my son here in my house. I couldn't go to the posta anymore."*– 25yrs, Home

Accessing care presented as a major barrier to accessing IPC for many women. Within Iquitos, transport is easily accessible however, within a couple of miles of the city, transport services become less reliable and regular.

## Theme 3—Fear of healthcare facilities

Throughout many of the interviews, women expressed fear and unfamiliarity of healthcare facilities.

*"I have been told [to give birth in hospital], but I was afraid because I've never been to the hospital"*– 28yrs, Posta

One woman feared attending hospital due to the death of a child in hospital.

*"I feel like having [the baby] in my house, umm because going to the hospital, I feel shy since I lost my baby"*– 39yrs, Home

In addition, participants feared a hospital birth in which interventions were required and autonomy was lost, including a fear of 'cutting' and a desire for a 'normal' vaginal birth.

*"I wanted to have my baby boy this way without cuts, but no, I couldn't. . . they had already done an evaluation for me, and I was not going to be able to have a natural, normal baby. They told me. . . they were going to do a C-section"*– 29yrs, EsSalud

One woman also discussed how women's partners wanted them to have a natural delivery with the fear of deserting the woman if she required an intervention. This stemmed from a desire for women to have the 'courage' for a natural birth.

*[Why didn't you want the C-section?] "Because, no one is safe in life, suddenly my husband leaves me"*– 39yrs, Home

As well as this, participants often discussed feeling uncomfortable with other aspects of care associated with healthcare facilities, including exposure to HCPs and a fear of 'touching'.

*"They stick their finger in and they say. . . if the baby is coming or is about to come. . . in the house, you just feel the baby coming, and you tell the person helping that it's already born. . . but in the hospital, after a while they touch you again."*– 18yrs, Home

This woman directly compared the care provided to women at hospital to home, with less intervention occurring at home. Another woman also reported feeling uncomfortable with repeated examinations and implied a discomfort towards male HCPs.

*"They started to check me inside . . . it turns out that the baby was in a different position. Lots of doctors came in, first one comes in, touches me, inserts his fingers in the vagina, touches. . . then another doctor comes in. . . another doctor does the same thing, a male comes in, also the same thing, and two. . . two more doctors come in, just the same."*– 29yrs, Hospital

As well as direct inferences to fear of hospitals, many women discussed their experiences of hospital care, which can have implications for others decisions. A handful of women had negative experiences in hospital, including poor attention from HCPs.

*"I don't know if it's because the hospital is big, they don't give you adequate attention, they don't listen to you"*– 44yrs, Posta

Women also reported poor treatment from HCPs.

*"Some obstetricians don't treat you properly. . . I had this experience that they yelled at me, this one [obstetrician], went crazy because it hurt a lot"*– 20yrs, Hospital

Experiences like this could feed into other women's fears of hospital care. Finally, a selection of women feared separation from their babies during hospital stays.

*"While she was in the hospital, I was uncomfortable because she was out of my sight and I was in the hospital as a sick person, and my little girl was in the incubator. I didn't see her."*– 39yrs, Home

Many of the participants experienced a fear of healthcare facilities or interventions and practices deterring them from accessing care. However, a handful of women also experienced poor attention and treatment in facilities which could prevent them and others accessing care in the future.

## Theme 4—Importance of seeking care

Throughout the interviews, women's awareness and knowledge of diseases in pregnancy and childbirth varied. Many women realised the importance of seeking IPC, discussing their fears of homebirths.

*"I have to be in a hospital because if I gave birth in a house or in some other place outside the hospital, my mother thought that suddenly something would happen to me, that is, she thought that suddenly I would die or my babies would"*– 21yrs, Hospital

Fear sometimes stemmed from stories they had heard about homebirths:

*"They arrived with their dead babies already. . . that's what I saw, so that's why I decided not to have it in my house. . . because it's going to happen to me. . . so I decided to go to hospital, for my safety and for my baby."*– 31yrs, Posta

Additionally, women talked about the availability of medication, equipment and HCPs in supporting their decision to give birth in healthcare facilities.

*"In an emergency, anything that you don't expect to happen, can happen in labour. . . the positive side is that you have specialists by your side, and they can take care of you. If you give birth at home, I think there is a higher death rate of pregnant women"*– 34yrs, Hospital

However, this awareness of risk was not universal, and a couple of women did not believe they were at risk when delivering at home.

*"My mum took care of all my sisters-in-law to have their babies at home, and that gave me the courage to have my baby here in my house because I knew nothing would go wrong."*– 18yrs, Home

Furthermore, other women who had homebirths believed that 'normal' antenatal appointments and a healthy pregnancy meant they did not need to attend hospital. Additionally, women believed that if the labour became complicated, they could then attend hospital.

*"If the baby is well, it's fine, if the baby is bad, we can go [to the hospital]."*– 39yrs, Home

However, the majority of women followed advice given at ANC appointments, particularly those with previously successful hospitals births, and attended healthcare facilities for their delivery.

*"When I was going through my pregnancy appointments, the gynaecologist and obstetrician told me, in an emergency we can go to any health centre, the closest thing for the wellbeing of me and my baby."*– 20yrs, EsSalud

## Theme 5—Comfort and traditions of home

The final theme generated from the data was the comfort of homebirths and traditions that prevented women seeking help. This included having loved ones around and the familiarity of home.

*"I'm better off at home, I'm not worried that I'm in another bed, I know. . . I'm being treated, they're giving me my warmth, my children, my husband."*– 39yrs, Home

As well as this, many women discussed the traditional remedies that were available at home, with some women mentioning these could not be taken into hospitals or postas.

*"My sister-in-law was making me [the hot drink], so that my baby comes out quickly."* [*What does it do?*] *"make the pain go away faster and to make the baby come out"*– 18yrs, Home

Additionally, all the woman who delivered at home discussed the normality of homebirths and due to previous successful home deliveries, they were more inclined to have another. Participants also discussed how after a homebirth it was possible for them to seek professional care by attending a posta within the following days for reassurance.

*"On the second day [after the birth] I go to the posta, they give him his vaccination and after 8 days I go for his check-ups."*– 39yrs, Home

Giving birth at home also meant for many women that they could continue with day-to-day activities.

*"I'm having my babies normally, I have to take medicine, do my laundry, that's all when you have them at home."*– 36yrs, Home

For many, a key part of this was the ability to care for their other children.

*"In the home sometimes. . . they don't have relatives who look after their children, and. . . that's why they decide to have them in their home, because so many things are being known in these times, there is rape. . . that's why I'm going to be thinking in the hospital, how is my daughter, how is my son?"*– 30yrs, Home

Finally, a couple of the women preferred giving birth at home for the autonomy it gave. As previously mentioned, women preferred a natural birth with the avoidance of interventions. Delivering at home also meant women had the ability to choose their birthing position.

*"It's possible for you to have your baby lying down or sitting down, or squatting. . . I had my baby sitting down. That's the difference. I mean, in the hospital, they make you lie down"*– 18yrs, Home

For many of the participants who gave birth at home, the comfort and familiarity played a large role in their decision making. This combined with other factors such as the cost and availability of transport and the fear of healthcare centres resulted in women choosing to have homebirths.

## Discussion

### Principal findings

Many barriers persist for women accessing IPC in Amazonian Peru, highlighting the complex and multifactorial nature of accessing SBAs in LMICs [14]. Despite free IPC provided by the government, many financial barriers were still cited. This included indirect costs of facility births, such as transport, and the additional costs of medication. Furthermore, the tradition of homebirths and fear of hospitals and interventions prevented women accessing care. Several other barriers were also identified including distance to healthcare facilities and fast onset of labour. Women who had delivered in healthcare facilities had mixed experiences, some discussing poor care, including poor attention and verbal abuse. Women's awareness of risks associated with childbirth were mixed; many seeking medical care for the safety of themselves and the new-born and others unaware of the importance of SBAs.

### Comparison with literature

Barriers within all aspects of Thaddeus and Maine's model were identified (Table 4). Many of the themes identified from the data confirms work from other settings, whilst adding new insight into barriers in rural Peru.

Financial barriers have been reported by women worldwide [21, 26, 32, 37, 59–61]. Due to the existence of the government funded SIS and EsSalud and recent literature from Loreto, the authors initially thought finances wouldn't be a key barrier [35]. However, costs, and the fear of hidden costs, were experienced by women including for transport and medication [62]. Several studies from other settings with free IPC also highlighted that women experienced similar issues [25, 27, 37, 63], including in Laos, where women were required to pay for medical equipment [31].

**Table 4. Summary of findings summarised within the Three Delays Model.**

| Aspect of the Three Delays Model | Finding |
| --- | --- |
| Phase 1 delays–Decision to seek care | Direct and indirect costs of healthcare facility birth |
| | Fear and unfamiliarity of healthcare facilities |
| | Previous healthcare experiences |
| | Fear of interventions in hospital |
| | Poor awareness of risks and diseases associated with pregnancy |
| | Familiarity of home |
| | Normality of homebirth |
| | Availability of IPC at home |
| | Interruption to daily activities and childcare |
| Phase 2 delays–Identifying and reaching medical facilities | Cost of transport |
| | Distance/time to healthcare facility |
| | Fear of delivering en route to healthcare facilities |
| | Unavailability of transport |
| | Onset of labour at night–lack of transport |
| | Reliance on public transport |
| | Speed and pain of labour |
| Phase 3 delays–Receipt of adequate and appropriate treatment | Poor attention from HCPs |
| | Verbal abuse from HCPs |
| | Unfamiliarity with hospital care |

Many women's birth location decisions were driven by fear; women delivering at home feared hospitals and interventions, whilst women choosing to deliver in healthcare facilities feared the risks for mother and new-born with homebirths. The fear of hospital births has been found in other studies, including beliefs that every woman attending hospital for childbirth receives 'cutting' [37, 64]. Women favoured a natural birth and it was noted that women's partners also preferred women to deliver naturally. To the authors' knowledge this has not been reported before and contrasts other publications such as Pazandeh et al., who reported that women in Iran feared their future sexual appeal and satisfaction of their husbands, following pelvic floor injury with a natural delivery [65].

A selection of women did not seek care due to poor knowledge of risks. This has previously been documented, including the belief that hospital care is only necessary when experiencing obstetric complications [20, 23, 27, 29, 37, 63, 64, 66, 67]. The final phase 1 barrier, the comforts of home, also agreed with many findings amongst the global literature. One aspect included the ability to be cared by family members, in contrast to hospitals where they may not be permitted in the room [29, 30, 32, 64]. Furthermore, a study from Burkina Faso found women were unable to take traditional drinks into hospitals [21]. Likewise, in the Peruvian Amazon, Westgard et al., found women feared prenatal vitamins and preferred traditional remedies [36]. Other studies have also found women prefer homebirths to enable continuation of daily activities, including care for other children [37, 61]. However, the fear for the safety of their children whilst at hospital has not previously been identified as a barrier. Finally, women preferred giving birth at home to allow choice in birthing position. This confirms other studies where women said in hospital, they have to be in a supine position [20, 21, 30, 32]. Women in Rural Northern Ghana also reported this; More flexibility in birthing positions were possible with traditional birth attendants who allowed any position, given it would not harm the mother or baby [32].

Similarly, to Iquitos, transport is often reported by women as a primary reason for not being able to seek IPC, including unavailability of transport at night and lack of suitable transport [26, 31, 32, 37, 59, 60, 64, 68, 69]. A selection of studies also found that women had transport issues dependent on seasonal rainfall. In contrast to Loreto, where the dry season caused difficulties, the rainy season, including flooding and landslides, caused accessibility problems for other women elsewhere [21, 28, 32, 70].

A selection of women experienced negative treatment in healthcare facilities including poor attention and verbal abuse. These abusive behaviours are not confined to women in this setting with a recent paper published reporting that 41.6% of observed women in 4 LMICs experienced some form of abuse, stigma or discrimination [71]. Other qualitative papers note that poor attention or treatment in healthcare facilities influences women's future birth location decisions, both in high- and low-income settings [21, 26, 32, 37, 72].

Whilst similar research is being conducted in other LMICs and women are encouraged to deliver in healthcare facilities, many high-income countries are now supporting women to deliver at home [73, 74]. For low-risk women, this is being shown to be a safe choice with reduced rates of interventions and complications [75, 76]. However, for this to be safe, women need access to trained midwives, a good referral system and reliable transport.

## Strengths and limitations

As far as the authors are aware, this is the first study to explore barriers to facility births in Amazonian Peru. All women recruited had given birth in the prior 18 months, improving participant's recall. Furthermore, through a diverse recruitment strategy, women who had delivered at healthcare facilities and home were included to ensure facilitators and barriers were

explored. The presence of a local interpreter ensured comfort for participants and accurate translations of local dialects. To reduce the likelihood of misinterpretation, both the English and Spanish from the recordings were transcribed and compared [77, 78].

Due to time restraints of the project, it was not possible to do respondent validation. To improve the analysis of data and increase credibility, analyst triangulation was performed, and the final results were discussed between the authors [41, 79]. Cultural differences are likely to have impacted the data however the researcher made attempts to remain unbiased and reflexive throughout the process, including de-briefing with the local supervisor and a reflexive diary. Furthermore, interviews were conducted in settings where participants were comfortable, and a local interpreter was always present.

## Implications

Despite the provision of free IPC in Peru, women continue to face financial barriers. Until these cost barriers are removed it is likely that difficulties will persist. Further research needs to be conducted into the additional costs incurred by those in the Peruvian Amazon across different fields of healthcare and methods to eliminate or reduce them. Increasing the provision of care in primary healthcare centres could help reduce transport costs and distance, helping to achieve UHC for childbirth. However, providing UHC, requires both the utilisation of care provided and good quality care [80, 81]; Research has shown that delivering in healthcare facilities does not always improve maternal outcomes [82–84]. A recent publication modelled a service delivery redesign to ensure women's outcomes were maximised, by encouraging women to deliver in larger, better equipped hospitals [85]. To enable access to better resourced facilities, maternity waiting homes (MWH) could be introduced. MWHs are residences located near hospitals, enabling access to obstetric care and removing the unpredictability of onset of labour, similarly to participants in this study who stayed with relatives prior to parturition [86]. Prior to their establishment, a "needs assessment" would be required to assess the health services available and geographical inaccessibility and acceptability to the local population [86]. It would also be important to consider potential barriers, including care for children and cost of travelling to MWHs [87]. MWHs have been established in other parts of Peru, for example Cuzco, where the Ministry of Health provided training to ensure culturally acceptable care and rates of homebirths have subsequently fallen [88]. Although MWHs may not remove all barriers to facility births, primarily financial, they would help to reduce problems associated with distance and the unpredictable nature of labour. Additionally, other methods to tackle travel expenses should be considered to address financial and geographical inequalities, for example travel cards for those who do not have access to personal transport or additional funds from the SIS [89]. These could also be used in conjunction with MWHs.

In addition to this, as a result of the negative care experiences, staff need to ensure women are treated with dignity and respect. This includes restricting the frequency of vaginal examinations [90]. Furthermore, changes to practice need to be made to alleviate fears about hospitals and interventions, including educating and reassuring women about the reasons for interventions. Additionally, as recommended by the WHO, women need to be given autonomy during childbirth, including birthing position [90]. To help reduce the cultural differences with care in hospitals and reduce fear, women should be allowed a birthing partner of choice [90, 91]. Ideally infrastructure would also be amended to ensure mothers and newborns remain together postpartum or allowing access to neonatal care units.

Further research should be targeted in other parts of the Peruvian Amazon, particularly further away from Iquitos where fewer facilities are available and accessing SBAs is more challenging. Furthermore, research should be conducted to establish how best to educate women

about the risks of childbirth and the importance of SBAs. This study can also aid the development of safe motherhood initiative and public health policies [92].

## Conclusion

Despite the WHO encouraging women to deliver in healthcare facilities with SBAs, women in the Peruvian Amazon continue to face barriers accessing IPC [10]. Several barriers found in this setting concur with the global literature including fear of hospitals, lack of transport and financial hurdles. However, barriers unique to this setting were also found; fear for children's safety whilst in hospital and a fear of caesareans, partly due partners leaving women if interventions are required. Women in Loreto have also experienced abusive behaviour in health facilities. Changes to practice and facilities are required, including changes to the attitude of staff and modifications to ensure mothers and babies remain together postpartum. Further research should be conducted to assess the suitability of MWHs in the region and explore women's barriers to IPC in other parts of Loreto.

## Supporting information

**S1 File. Consolidated criteria for reporting qualitative studies (COREQ): 32-item checklist.**
(PDF)

**S2 File. Visualisation of the recruitment process.**
(PDF)

**S3 File. Summary of topic guide.**
(PDF)

**S4 File. Themes, subthemes & codes breakdown.**
(DOCX)

## Acknowledgments

We thank the healthcare staff in primary healthcare centres in Iquitos for helping in the recruitment of participants and the interpreter for helping with the conduction of interviews and transcription. Secondly, we would like to thank the women who gave their time to be involved in the research and share their experiences and beliefs.

## Author Contributions

**Conceptualization:** Esme Gardiner, Graciella Meza, Gilles de Wildt, Beck Taylor.

**Data curation:** Esme Gardiner, Beck Taylor.

**Formal analysis:** Esme Gardiner, Beck Taylor.

**Funding acquisition:** Esme Gardiner.

**Investigation:** Esme Gardiner.

**Methodology:** Esme Gardiner, Gilles de Wildt, Beck Taylor.

**Project administration:** Esme Gardiner, Graciella Meza, Gilles de Wildt.

**Resources:** Esme Gardiner, Graciella Meza.

**Software:** Esme Gardiner.

**Supervision:** Graciella Meza, Gilles de Wildt, Beck Taylor.

**Validation:** Jo Freda Lai, Divya Khanna, Beck Taylor.

**Visualization:** Esme Gardiner.

**Writing – original draft:** Esme Gardiner.

**Writing – review & editing:** Esme Gardiner, Jo Freda Lai, Divya Khanna, Graciella Meza, Gilles de Wildt, Beck Taylor.

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
