## [Decision Letter · Decision Letter 0]

9 Mar 2021

PONE-D-20-36773

Exploring women’s decisions of where to give birth in the Peruvian Amazon; Why do women continue to give birth at home? A qualitative study

PLOS ONE

Dear Dr. Gardiner,

Thank you for submitting your manuscript to PLOS ONE. After careful consideration, we feel that it has merit but does not fully meet PLOS ONE’s publication criteria as it currently stands. Therefore, we invite you to submit a revised version of the manuscript that addresses the points raised during the review process.

We look forward to receiving your revised manuscript.

Kind regards,

Juliet Kiguli, MA, PhD

Academic Editor

PLOS ONE

Additional Editor Comments:

Dear Authors, Please address all comments and indicate a thematic framework for the analysis.

Journal Requirements:

2. Please include additional information regarding the survey or questionnaire used in the study and ensure that you have provided sufficient details that others could replicate the analyses. For instance, if you developed a questionnaire as part of this study and it is not under a copyright more restrictive than CC-BY, please include a copy, in both the original language as well as the English version already provided, as Supporting Information.

5. We note that Figure 2 in your submission contain map images which may be copyrighted. All PLOS content is published under the Creative Commons Attribution License (CC BY 4.0), which means that the manuscript, images, and Supporting Information files will be freely available online, and any third party is permitted to access, download, copy, distribute, and use these materials in any way, even commercially, with proper attribution. For these reasons, we cannot publish previously copyrighted maps or satellite images created using proprietary data, such as Google software (Google Maps, Street View, and Earth). For more information, see our copyright guidelines: http://journals.plos.org/plosone/s/licenses-and-copyright.

5.1.    You may seek permission from the original copyright holder of Figure 2 to publish the content specifically under the CC BY 4.0 license. 

5.2.    If you are unable to obtain permission from the original copyright holder to publish these figures under the CC BY 4.0 license or if the copyright holder’s requirements are incompatible with the CC BY 4.0 license, please either i) remove the figure or ii) supply a replacement figure that complies with the CC BY 4.0 license. Please check copyright information on all replacement figures and update the figure caption with source information. If applicable, please specify in the figure caption text when a figure is similar but not identical to the original image and is therefore for illustrative purposes only.

Reviewers' comments:

Reviewer's Responses to Questions

**Comments to the Author**

1. Is the manuscript technically sound, and do the data support the conclusions?

Reviewer #1: Yes

Reviewer #2: No

2. Has the statistical analysis been performed appropriately and rigorously? 

Reviewer #1: N/A

Reviewer #2: N/A

3. Have the authors made all data underlying the findings in their manuscript fully available?

Reviewer #1: No

Reviewer #2: No

4. Is the manuscript presented in an intelligible fashion and written in standard English?

Reviewer #1: Yes

Reviewer #2: Yes

5. Review Comments to the Author

Reviewer #1: This is an interesting paper on a study carried out in an Amazonian setting, not commonly the locus for studies of women's experience of pregnancy and childbirth.

The background is generally appropriate concerning MDGs and maternal mortality rates in a range of contexts including LMICS. However, from the start a causal relationship is assumed in linking the proportion of home births in the region directly with the higher MMR. Care needs to be taken in this regard as other factors may be contributing such as those raised later in the discussion section.

It would have been useful to have description for the reader of how maternity is organized in Peru and in the region in which the study took place- large and small hospitals, clinics and so on. Also the way that geography impacts e.g. seasonal variation in accessibility and the way that the healthcare system addresses that point.

The methods are largely appropriate for this qualitative study. However, the Topic Guide seems a rather highly structured one, with many questions used. It seems more like a structured interview, not necessarily supporting women's views in directly reflecting on their life experience of childbirth and maternity care, particularly intrapartum and the decisions they and others made. This is a limitation and should be mentioned in the appropriate discussion section. I note that the interviews were relatively short (on average 34 minutes - the range could be added).

The authors describe mostly anticipated themes as reflected in the structure of the topic guide. However, the unexpected theme relating to care of their other children is of interest and has been described in some studies of Aboriginal and Torres Strait Islanders. That women and their partners favoured a natural birth is of particular interest and not given sufficient weight in the discussion. Fear concerning the medicalization of birth and excessive examinations during labour are however recognised as a concern that needs to be addressed if what is reported in this qualitative study is reflected on a larger scale.

The suggestions for changes in the organization of maternity care are of interest, though do not necessarily arise from the study data and that should be clear in the write up.

A minor point concerns the repetition of 'a selection of...' - women, studies, papers.

Reviewer #2: Dear editor

Many thanks for the opportunity for reviewing this manuscript. This topic is important as it is related to seeking childbirth care in health facilities under skilled birth attendants' supervision.

Methods: The authors reported that this is a descriptive qualitative study that is used thematic analysis. However, they only have provided some explanations under each theme using some quotations from the participants. It seems that analysis is too superficial and is in the very early stage. There is a need to use sub-themes under each theme and some information about the rigour of the study.

Results: The results show that the authors have reported five themes, and there is no information about the subthemes. Additionally, the explanations written under the themes are not well organised, and some of them could be moved to other themes. The data shows an overarching theme linked to disrespect and abuse during labour and childbirth, which is the main barrier that women do not seek care in the facilities (The suggestions are mentioned in the text that authors may use for more in-depth analysis).

To better understand the results, I recommend using a table of the codes, sub-themes, and themes.

Discussions: The first paragraph is repeating the results, and the rest of the discussion is fragmented and needs to be organised. I suggest writing a new discussion based on the revised results linked to the themes and sub-themes.

6. PLOS authors have the option to publish the peer review history of their article (what does this mean?). If published, this will include your full peer review and any attached files.

Reviewer #1: **Yes: **Maggie Redshaw

Reviewer #2: **Yes: **Farzaneh Pazandeh

---

## [Author Response · Author response to Decision Letter 0]

26 Jul 2021

The following has also been provided as an attached file, named 'Response to Reviewers'.

Dear Dr Kiguli, 

Response to review: Exploring women’s decisions of where to give birth in the Peruvian Amazon; Why do women continue to give birth at home? A qualitative study

We thank you and the Reviewers for your comments. We have addressed each recommendation systematically, explained in the tables overleaf, and highlighted in the attached manuscript. In addition, we make the following statements:

• Authors: Two of the authors (Jo F Lai & Graciela Meza) have had their names updated on the author list on the Editorial Manager to reflect their preferences and align with other publications (Jo Freda Lai & Graciela Rocio Meza Sanchez)

• Funding Information: [We would like to confirm the following statement as required by the PLOS One Submission Guidelines]. “The funders had no role in study design, data collection and analysis, decision to publish, or preparation of the manuscript.” 

• Ethics Statement: [The Ethics Statement has been updated to include the reference numbers]. “Written ethical approval was provided locally by the Regional Directorate of Health, Loreto (385-2019-GRL-DRSL/30.09.01) and The University of Birmingham – BMedSc Population Sciences and Humanities Internal Research Ethics Committee (IREC2019/1548737). All participants provided written, informed consent prior to conducting the interviews.”

• Copyrighted figures: Figure 2 is a map illustrating the recruitment sites for participants. This is a modified copy of a map provided by OpenStreetMap. OpenStreetMap is a site which allows the use of its maps under the open database license CC BY-SA. Both the image and figure title fully credit the source, following their published guidelines. [Please see https://www.openstreetmap.org/copyright ] 

• Data Availability: This study is based on a dataset of 25 qualitative interview transcripts. However, the authors did not seek ethical permission from the participants, nor the ethics committee, for the data to be used for anything other than this particular research study. The authors therefore do not have explicit permission for data sharing, re-analysis nor future studies and so would be inappropriate and unethical to make them available in the public domain. Furthermore, the data contains potentially identifying patient information. However, qualified individuals can direct queries by contacting Dr Ruth Riley (r.riley@bham.ac.uk) - Chair of the University of Birmingham BMedSci Intercalation Internal Ethics Review Committee.

Thank you for your consideration and we look forward to hearing from you,

Yours sincerely 

Miss Esme Gardiner

Journal Requirement 

Please address all comments and indicate a thematic framework for the analysis. Thank you for highlighting the need for more clarity regarding the analysis. We have indicated more clearly in the methodology that Braun and Clarke’s Thematic Analysis was used (page 8 – line 196). Furthermore, we have detailed more clearly the process of the analysis throughout the data analysis section of the methodology (Page 8-9). 

Journal Requirement 1

Please ensure that your manuscript meets PLOS ONE's style requirements, including those for file naming. To the best of our knowledge, the manuscript meets the journal’s style requirements, including the figures and supplementary file names. 

Journal Requirement 2

Please include additional information regarding the survey or questionnaire used in the study and ensure that you have provided sufficient details that others could replicate the analyses. We would like to clarify that the topic guide has been provided as a supplementary file (supplementary file 3). In addition to this, the methodology has been modified to clarify the process of analysis (see above). This has included putting the process into a clear chronological order and documenting the co-authors contribution to the analysis (page 8-9). 

Journal Requirement 3

We note that the grant information you provided in the ‘Funding Information’ and ‘Financial Disclosure’ sections do not match.

When you resubmit, please ensure that you provide the correct grant numbers for the awards you received for your study in the ‘Funding Information’ section. The Editorial Manager and manuscript have been updated to ensure that the Funding Information and Financial disclosure match. This included funding from the Topham Bursary Fund (which no award number was provided for) and the University of Birmingham Medical and Dental College who covered the expenses of the research trip. 

In line with the manuscript style requirements, the funding bodies have been removed from the acknowledgements. 

We have been unable to find the required space on the editorial manager to confirm and add the sentence “The funders had no role in study design, data collection and analysis, decision to publish, or preparation of the manuscript.” – As stated on the PLOS One submission guidelines. 

Journal Requirement 4

In your Data Availability statement, you have not specified where the minimal data set underlying the results described in your manuscript can be found. […]

Important: If there are ethical or legal restrictions to sharing your data publicly, please explain these restrictions in detail. Due to ethical restrictions, it would not be possible to place the interview transcripts into the public domain. Ethical approval was sought from the University of Birmingham and locally in Loreto (both ethics reference numbers provided) in which ethical approval was not obtained to share the transcripts outside of the research team. The transcripts contain potentially identifiable information for participants. Details have been given for Ruth Riley, Chair of the ethics board used by The University of Birmingham, for direct enquiries. This is also reflected by a paper published by PLOS One which utilised the same Ethics Board (https://doi.org/10.1371/journal.pone.0209736) 

Journal Requirement 5 

We note that Figure 2 in your submission contain map images which may be copyrighted. 

 Following an initial enquiry to Google Maps, it was decided to change the source to OpenStreetMap, in line with journal requirement 5.2. OpenStreetMap is a site which allows the use of its maps under the open database license. Both the image and figure title fully credit the source, following their published guidelines. The reference has also been updated. 

Reviewers’ comments Action/comments

Reviewers’ Responses to Questions 

1. Is the manuscript technically sound, and do the data support the conclusions?

Review 1 – Y 

Reviewer 2 – N Following the adjustments and comments made below, we believe we have satisfied Reviewer 2’s suggestions. 

Reviewers’ Responses to Questions 

2. Has the statistical analysis been performed appropriately and rigorously?

Review 1 – N/A

Reviewer 2 – N/A No amendments required 

Reviewers’ Responses to Questions 

3. Have the authors made all data underlying the findings in their manuscript fully available?

Review 1 – N

Reviewer 2 – N This relates to journal requirement 4, please see above. It is not possible to release the transcripts into the public domain, however this is in line with the ethical approval obtained and for the participants’ privacy.

Reviewers’ Responses to Questions 

4. Is the manuscript presented in an intelligible fashion and written in standard English?

Review 1 – Y 

Reviewer 2 – Y No amendments required 

Reviewer 1 – Comment 1 

The background is generally appropriate concerning MDGs and maternal mortality rates in a range of contexts including LMICS. However, from the start a causal relationship is assumed in linking the proportion of home births in the region directly with the higher MMR. Care needs to be taken in this regard as other factors may be contributing such as those raised later in the discussion section. Changes have been made in the introduction to ensure that it is clear that this causative assumption has not been made (Page 3, lines 76-78). 

Reviewer 1 – Comment 2

It would have been useful to have description for the reader of how maternity is organized in Peru and in the region in which the study took place- large and small hospitals, clinics and so on. Also the way that geography impacts e.g. seasonal variation in accessibility and the way that the healthcare system addresses that point.

 We are grateful to Reviewer 1 for highlighting this gap in the introduction and recognise the value that this adds. As a result, we have added more detail generally about the healthcare system in Peru. This focusses on the funding of the system, whereby it is not strictly a universal healthcare system with multiple organisations funding different groups of people, allowing access to different aspects of the healthcare system (page 3-4, lines 87-104). As well as this, we have further detailed information about the location, provision and care providers of obstetric care. Very little information exists about the formal structure of maternity care in Peru, so some of this information is collated from the principal researcher’s experiences and Dr Meza, a co-author who is based in Iquitos. In addition to this, information has been added from the World Health Organisation and a paper produced from research locally about poor maternity provisions (DOI: 10.4269/ajtmh.14-0536). 

Reviewer 1 – Comment 3

The methods are largely appropriate for this qualitative study. However, the Topic Guide seems a rather highly structured one, with many questions used. It seems more like a structured interview, not necessarily supporting women's views in directly reflecting on their life experience of childbirth and maternity care, particularly intrapartum and the decisions they and others made. This is a limitation and should be mentioned in the appropriate discussion section. Thank you for highlighting this issue. We would like to confirm that although the topic guide may appear to be highly structured, it was used as a framework for discussion, and participants were encouraged to share their views and experiences. The guide was used with flexibility. Furthermore, the topic guide was refined with data collection, reflecting the iterative nature of this project. 

It was useful for us at times to have a more-structured topic guide as occasionally women were less able to or willing to talk extensively without prompting. 

We have mentioned this in the methodology (line 176) and reflected on this as a possible limitation to the study in the discussion (lines 670-675). 

Reviewer 1 – Comment 4 

I note that the interviews were relatively short (on average 34 minutes - the range could be added).

 We have added the range to the interviews (line 169) as we agree this would be helpful for readers. The shortest interview was much shorter than the remainder as the participant had very little to say and appeared to be disengaged, however they did not want to withdraw from the study. We also reflected on this at the time and considered it to be possibly partially due to our recruitment strategy. Interviews were not pre-organised, instead women recruited at the healthcare centres were offered to have their interview on the day, often leading to them spending far longer at the centres than they may have anticipated. 

Reviewer 1 – Comment 5 

The authors describe mostly anticipated themes as reflected in the structure of the topic guide. However, the unexpected theme relating to care of their other children is of interest and has been described in some studies of Aboriginal and Torres Strait Islanders. That women and their partners favoured a natural birth is of particular interest and not given sufficient weight in the discussion. Thank you for suggesting this. We have reviewed the literature about Aboriginal and Torres Strait Islander’s experience and have incorporated this into our discussion (line 545) - DOI: 10.1016/j.wombi.2016.01.004. For the finding concerning men preferring women to give birth vaginally, we have expanded this within our discussion and suggested involving women’s partners in antenatal training. 

Reviewer 1 – Comment 6 

Fear concerning the medicalization of birth and excessive examinations during labour are however recognised as a concern that needs to be addressed if what is reported in this qualitative study is reflected on a larger scale. We agree that fear surrounding childbirth in healthcare settings is a major problem in lower- and middle-income countries as highlighted by the recent Lancet paper. Therefore, we have taken this comment on board and added this to the implications section, including that about excessive examinations (Lines 715-725). 

Reviewer 1 – Comment 7 

The suggestions for changes in the organization of maternity care are of interest, though do not necessarily arise from the study data and that should be clear in the write up.

 Thank you for mentioning this, we agree that it was not initially clear how the implications aligned with the results. Following reviewer 2’s comments, we have clarified the links from the results to the discussion/comparison with the literature to highlight the change in order from our themes to the findings in relation to The Three Delays Model and the global literature. We have also tied this through to the implications with the addition of several paragraph indentations to more clearly separate the suggestions. Furthermore, several sentenced have been added to help clarify where the implications relate to the data. For example line 703-706: “It would also be important to consider potential barriers, including care for children and cost of travelling to MWHs (91). This is particularly important considering cost of transport and fear for other children’s safety whilst in hospital were found to be barriers to IPC.”

Reviewer 1 – Comment 8 

A minor point concerns the repetition of 'a selection of...' - women, studies, papers. Reviewed and amended to reduce repetition of phrase. 

Reviewer 2 – Comment 1 

The authors reported that this is a descriptive qualitative study that is used thematic analysis. However, they only have provided some explanations under each theme using some quotations from the participants. It seems that analysis is too superficial and is in the very early stage. There is a need to use sub-themes under each theme and some information about the rigour of the study. Thank you for clarifying the need for more depth within the methodology. We have added a more detailed chronological description of the analysis, in line with Braun and Clarke’s 6 step guide to thematic analysis (lines 196-204). This more clearly describes the process from familiarisation, the generation of a code book through to subthemes and themes. Subthemes are detailed later on in table 3 (page 9). 

In addition, we have added greater detail about steps taken to improve the rigour of the study (lines 658-665). This has been added to the strengths and limitations section of the discussion. 

Reviewer 2 – Comment 2 

The results show that the authors have reported five themes, and there is no information about the subthemes. Additionally, the explanations written under the themes are not well organised, and some of them could be moved to other themes. 

 We have, in line with comment 4, added a supplementary file (file 4) detailing the themes, subthemes and codes as well as a description of each theme. We decided to add this as a supplementary file in addition to table 3 for brevity for the reader, however this resource is now available to readers to support the main manuscript. 

The analysis and decisions on the themes and subthemes were discussed within the research team at all stages of the analysis. We do acknowledge that there is overlap and this is recognised within the results section of the manuscript. For example, transport costs are relevant to both Theme 1 (financial barriers) and theme 2 (accessing care). However, we discussed these codes and subthemes in detail as a team and allocated them to where was deemed most suitable. They are also briefly recognised within the results section (e.g., lines: 274, 531, 542) however are not repeated under every theme for brevity. 

To improve the results section, we have also expanded on the explanations surrounding both the themes generally, with a summarising paragraph at the end of each, and within the themes themselves to support the quotations. 

Reviewer 2 – Comment 3

The data shows an overarching theme linked to disrespect and abuse during labour and childbirth, which is the main barrier that women do not seek care in the facilities (The suggestions are mentioned in the text that authors may use for more in-depth analysis).

 We have discussed this thoroughly as a team, and revisited our analysis and interpretation, but while this is a major global issue, we have not identified this as an overarching theme in this study. This is a key challenge in Low- and Middle-Income Countries, as illustrated by the recent Lancet Paper and we have highlighted the need for further research to identify whether this is more of a system-wide problem. We believe the barriers to IPC, in Loreto, to be multi-factorial, as highlighted by Thaddeus and Maine’s 3 delays Model. If there were to an underpinning finding, we would believe this topic to be fear, for example, fear of healthcare facilities, fear of interventions, fear of delivering en route and fear of homebirths. We have therefore made this clearer at the start of the discussion within the principal findings (lines 564-573). We can confirm that this does not introduce any new findings or concepts into the discussion however allows a series of findings to be joined by their common subject of fear. 

Reviewer 2 – Comment 4 

To better understand the results, I recommend using a table of the codes, sub-themes, and themes.

 Thank you for highlighting this. We have amended the manuscript to illustrate the results in a clearer manner. We have followed advice and added a supplementary file detailing the themes, subthemes and codes as well as a description of the theme (supplementary file 4). Furthermore, we have added more detail to the descriptions surrounding the quotations and added a summarising paragraph at the end of each theme within the results section (lines: 265-270, 317-323, 409-415, 474-482 and 539-544). 

 Additionally, within the introduction of the results we have indicated (line 208) that the results follow the structure of the themes/subthemes table (table 3) to help guide the reader. 

Indicated in the introduction of the results that they follow the structure of the themes/subthemes table

Finally, we have also added the theme numbers to table 4. This should help clarify where our findings sit within the Three Delays Model. 

Reviewer 2 – Comment 5 

The first paragraph is repeating the results, … We have reflected on this and cut down the paragraph to avoid repetition. We decided to keep a brief summary, as some readers may provide this helpful and it aligns with other PLOS One papers (see below). 

We also added the paragraph relating to our key finding of fear (from comment 3). As fear arose within multiple of the themes, tying them together and illustrating a common backbone to many of the barriers to IPC, we wanted to clearly bring this findings together for the reader. 

• Marsland et al., 2019 - https://doi.org/10.1371/journal.pone.0209736

• Bohren et al., 2015 https://doi.org/10.1371/journal.pmed.1001847

• Dodzo et al., 2017 https://doi.org/10.1371/journal.pone.0181771

Reviewer 2 – Comment 6 

… and the rest of the discussion is fragmented and needs to be organised. I suggest writing a new discussion based on the revised results linked to the themes and sub-themes.

 We have revised the results section to help organise and improve it for the reader (as detailed above). We have clarified that the discussion is aligned with The Three Delays Model (See table 4, and subheadings within the discussion – lines: 518, 555, 562, 571). We have presented our findings within table 4 at the start of the discussion which places the varying subthemes and findings within the model to help align it within the global literature. To improve clarity, following the findings, we have added the various themes that they fall under (Table 4 – Page 21-22). In addition to this, we have added titles to the discussion to indicate the relation to the Three Delays Model.

---

## [Editor Report · Decision Letter 1]

25 Aug 2021

Exploring women’s decisions of where to give birth in the Peruvian Amazon; Why do women continue to give birth at home? A qualitative study

PONE-D-20-36773R1

Dear Dr. Gardiner,

We’re pleased to inform you that your manuscript has been judged scientifically suitable for publication and will be formally accepted for publication once it meets all outstanding technical requirements.

Kind regards,

Juliet Kiguli, MA, PhD

Academic Editor

PLOS ONE
---

## [Editor Report · Acceptance letter]

31 Aug 2021

PONE-D-20-36773R1 

Exploring women’s decisions of where to give birth in the Peruvian Amazon; Why do women continue to give birth at home? A qualitative study 

Dear Dr. Gardiner:

I'm pleased to inform you that your manuscript has been deemed suitable for publication in PLOS ONE. Congratulations! Your manuscript is now with our production department. 

Kind regards, 

on behalf of

Dr. Juliet Kiguli 

Academic Editor

PLOS ONE